# Denonvilliers’ Fascia: The Prostate Border to the Outside World

**DOI:** 10.3390/cancers14030688

**Published:** 2022-01-29

**Authors:** Lazaros Tzelves, Vassilis Protogerou, Ioannis Varkarakis

**Affiliations:** 12nd Department of Urology, National and Kapodistrian University of Athens, Sismanogleion Hospital, 11526 Athens, Greece; ivarkarakis@med.uoa.gr; 2Department of Anatomy, School of Medicine, National and Kapodistrian University of Athens, Mikras Asias 21 St., 12462 Athens, Greece; vprotoger@med.uoa.gr; 33rd Urological Department, Attikon University Hospital, School of Medicine, National and Kapodistrian University of Athens, Rimini 1, 12462 Athens, Greece

**Keywords:** Denonvilliers’ fascia, prostate anatomy, radical prostatectomy, nerve sparing

## Abstract

**Simple Summary:**

Prostate cancer is a very common neoplasm in men, with surgery being a valuable tool for its successful management. The prostate gland lies deep in the male pelvis with several sheets of fibrous membranes surrounding it along anterior, lateral, and posterior surfaces. These membranes are called fasciae. Arteries, veins, and nerve fibers that are important for erectile function and continence can be found within these fasciae. An important fascia covering the posterior surface of the prostate and separating it from the rectum is Denonvilliers’ fascia. This structure is important for the confinement of cancer within the prostate and for completing an operation without damaging the nerves responsible for erectile function and continence while also removing all neoplastic tissue. This review covers the anatomical aspects of this structure, along with providing some clinical insight on how to use this knowledge to improve clinical outcomes.

**Abstract:**

The fascial structure around the prostate has been a controversial issue for several decades, but its role in radical prostatectomy is crucial to achieving successful nerve-sparing surgery. One of the fasciae surrounding the prostate is Denonvilliers’ fascia, forming its posterior border with the rectum and enclosing along its layers several fibers of the neurovascular bundle. This review focuses on embryological and anatomical points of Denonvilliers’ fascia, aiming to provide a summary for the operating general surgeons and urologists of this area.

## 1. Introduction

Prostate cancer (PCa) is the second most common neoplasm in men, comprising 15% of cancer diagnoses worldwide in 2012 [1]. Radical prostatectomy (RP) is a surgical procedure to remove the prostate gland with its capsule, seminal vesicles (SVs), and part of the ejaculatory ducts. It is commonly performed with curative intent for localized disease. Although a number of surgical techniques currently exist for RP, including open, laparoscopic, and robotic approaches, in the past, RP was associated with high mortality and morbidity rates. The pioneering findings regarding pelvic anatomy and splanchnic nerves responsible for erectile function, reported by Walsh and Donker back in 1982 using specimens from male fetuses or stillborn neonates [2] and adults [3], helped to improve impotence rates [4]. A number of studies regarding the anatomical structure of the male pelvis have been performed since, indicating that the individual patterns of pelvic nerve anatomy can significantly differ between patients and impact the surgical, functional, and oncological outcomes after RP.

Another common anatomical controversy regarding nerve-protecting surgery for prostate cancer concerns the fascial structure, nomenclature, and proper approach for dissection. Magnification offered by laparoscopic and robotic platforms showed that the lateral pelvic fascia and Denonvilliers’ fascia (DF) are potentially not single-layered structures but are composed of multiple sheets of tissue. This fact allows urologists to reconsider the pelvic anatomy and establish a variety of surgical techniques, such as the intra-, inter-, and extrafascial approaches for nerve-sparing surgeries [5].

Denonvilliers’ fascia is one of the fascial components that surround the prostate gland, along with the prostatic capsule and lateral or endopelvic fascia. Historically, this collagenous tissue was identified in 1836 by Charles-Pierre Denonvillier between the rectum and prostate, and was initially named the “prostato-peritoneal membrane”, based on the dissection of 12 male cadavers [6,7]. Although nearly 200 years have passed since this observation, many controversies still exist regarding both the embryologic origin and the detailed anatomy in adults. The dissection of the posterior surface and base of the prostate, where DF is located, is an important step in RP both for oncological and functional outcomes. The aim of this review is to present the anatomy of this fascia and provide insights for specific steps during RP.

## 2. Embryological Origin of Denonvilliers’ Fascia

Surgeons should be familiar with the embryological origin of structures in their surgical field as this helps to not only understand normal and abnormal development [8] but also to recognize the surgical anatomical planes and appropriate approaches. An ongoing scientific debate exists regarding DF, with three main existing theories. Cuneo and Vean were the first to describe the peritoneal fusion hypothesis, which implied that DF is formed after fusion of the two peritoneal folds in the region of the rectovesical (in males) or rectouterine (in females) cul-de-sac during embryonic development [9]. Smith et al. favored this theory with their anatomical studies in adults [10] and fetuses [11].

Later, Wesson et al. conflicted these findings by stating that the two peritoneal folds gradually disappear and cells with mesenchymal origin migrate to this location, multiply, and finally form DF [12]. This is the mesenchymal condensation theory [13]. The fusion theory was supported by Tobin and Benjamin, who performed dissection studies in embryos at various developmental stages [14]. They reported histological proof because they observed the existence of mesenchymal tissue between the rectum and genitourinary tract organs, which finally developed into muscle or connective tissue, such as DF [14]. The most persuasive evidence for supporting fusion theory was the discovery of mesothelium beside this mesenchymal tissue, which was thought to be a continuation of the embryonic cul-de-sac, and which later faded away, leaving mesenchymal tissue to form DF [14]. In a similar manner, the white line of Toldt was developed to form the lateral borders of the intestinal peritoneum [15]. A common misconception regarding these findings is that rectal fascia propria is the posterior layer of DF, even though the embryological origin of these two structures is different [16]. Diagnosis of cystic mesothelioma behind the prostate further supported the peritoneal fusion theory, as this case designated the existence of tissue with mesothelial origin to the anatomical location of DF [17].

A more recent theory was proposed by Kim who supported that DF is formed after mechanical pressure is applied to mesenchymal cells from the growing rectum, prostate, and SVs [18]. They observed that no mesenchymal cells were arranged to form a fascia and there was no peritoneal fusion, but instead proposed that cul-de-sac peritoneal folds gradually regress upwards at later stage embryos [18].

## 3. Surgical Anatomy

### 3.1. Pelvic Fascia Compartments

Fasciae are composed of connective tissue and cover not only muscles but also glands and vessels, although muscular fasciae are more easily identified compared to non-muscular fasciae [19]. As fascia identification during surgery provides planes for proper anatomical dissection in order to respect surgical margins and delicate nervous tissues or vessels, it is important for urologists to be familiar with their anatomy. The main fasciae covering the male pelvis are endopelvic fascia, Denonvilliers’ fascia, and prostatic fascia or capsule.

### 3.2. Prostatic Capsule

Whether or not the prostate is covered by its own capsule, meaning a dense layer of connective tissue, is controversial. Brooks et al. support that there is a capsule composed of elastin, collagen, and smooth muscle layers, which comes in continuity with endopelvic fascia in the anterolateral portion of the prostate and DF along the posterior surface [20]. Ayala et al. designed an important histopathological study where they examined prostate gland specimens for the existence of the prostatic capsule [21]. They concluded that the prostatic fibromuscular stroma is denser at the outer surface of the gland, forming a “pseudocapsule” rather than a true capsule, which is not continuous but is incomplete, especially at the apex [21]. Sattar et al. reached a similar conclusion with their morphometric analysis and reported that the prostatic capsule is formed as an extension of the parenchymal muscle layer, with a thickness ranging between 0.5 and 2 mm [22]. Lollo et al. suggested that no capsule exists but that organs are separated by loose connective tissue found between the prostate, SVs, and urinary bladder [23,24], while Young et al. did not detect any correlation between the frequently shown capsule in ultrasonic images and histopathological examination [25]. However, Kiyoshima et al., in their histopathological analysis, mention fibromuscular stroma and the prostatic capsule as different structures, while in the anterior part of the gland, they also did not detect a capsule but instead a more dense fibromuscular stroma [26]. In addition, they recognized the transition of the capsule to fibromuscular stroma across the lateral surface of the prostate [26].

## 4. Endopelvic Fascia (EPF)

Although EPF is a well-recognized and widely accepted anatomic structure, its nomenclature shows great heterogeneity, leading to misconceptions in the literature. As its anatomical variations strongly correlate with the achievement of nerve-sparing surgery, both in open and minimally invasive approaches, we provide a brief description.

Several synonyms exist in the literature to describe this fascia, including lateral pelvic fascia, superior pelvic fascia, endopelvic fascia, parietal fascia, levator fascia, parapelvic fascia, and periprostatic fascia [24]. Tewari et al refer to two distinct layers of periprostatic fascia: the prostatic fascia medially surrounding the prostate and the lateral pelvic fascia that overlies the levator ani muscle [27]. Kourambas et al., on the other hand, collectively name the total connective tissue between the prostate and levator ani as the lateral pelvic fascia [24,28]. Another widely used name is provided by Brooks et al. who refer to EPF as the fascia attaching to the levator ani muscle [20].

EPF covers the musculature of the pelvis (levator ani, piriformis, coccygeus, obturator internus) while it is a continuation of the transversalis fascia [24]. It forms the “fascial tendinous arch” across the lateral surface of the urinary bladder, as described by Myers [29]. EPF represents the lateral border of the lesser pelvis, as nerves and vessels leave the pelvis course along its outer border while pelvic structures course medially [24].

Initially, EPF was considered as a single-layered fascia, but the magnification achieved with advances in laparoscopic surgery shed light on the existence of multiple layers within this fascia and have permitted the development of intra-, inter-, and extrafascial dissection [30] (Figure 1 and Figure 2). Kiyoshima et al. found that in 48% of 79 radical prostatectomy specimens, the EPF and prostatic capsule were fused together without adipose or connective tissue between them [26]. In the remaining 52% of specimens, the levator fascia was separated from the prostatic capsule by varying the amounts of adipose tissue during their course [26]. Another histopathological study performed both in male fetuses and adults revealed that the levator fascia in men consists of multiple layers of elastic and smooth muscle fibers coursing in various directions [31].

## 5. Denonvilliers’ Fascia

Initially identified by Charles Denonvillier back in 1836 [6,7], this fascia represents the posterior border of the prostate and plays a significant clinical role because it is the barrier for PCa extension to the rectum and shows a close anatomical relationship with the neurovascular bundle (NVB), endopelvic fascia, rectum, and posterior surface of the prostate. It is very aptly described as the “plane between wind and water” by Ger [32].

The number of layers comprising DF has been a matter of discussion. Initially, it was considered a single-layered, connective tissue structure by C. Denonvillier. According to embryological dissection studies performed to demystify the embryological origin of DF, Smith [11] and Wesson [12] suggested the existence of two layers. Benjamin and Tobin later referred to the posterior layer of DF, naming it the rectal propria fascia, leading to confusion because these two structures have different embryologic origins [14].

This controversial issue was also examined by Kourambas et al. who performed an autopsy of adult male cadavers and studied the histological features of DF as well as its relationship with surrounding structures [28]. They support the existence of a single-layered fascia, which lies behind the prostate, in continuity with the pararectal fascia dorsally and the EPF ventrally, forming an “H” configuration [28]. In this study, the authors pointed out the existence of nerve fibers not only at the postero-lateral prostatic surface at the point of junction between DF and EPF/pararectal fascia but also at the midline part of DF, although at a smaller density [28]. On the same matter, Lindsey et al., after reviewing the existing literature, concluded that DF has no apparent separate layers, but there is a posterior layer, which corresponds to the rectal propria fascia [16]. The posterior layer of DF is also described by Nano et al. who refer to an existing space between the two DF layers, with the posterior closer to the rectum and anterior to the prostate [33]. Kinugasa et al., during their histological study on ten male cadavers, reported that DF consists of two or three laminae across the postero-lateral edge of the prostate, surrounding the NVBs and hypogastric nerves and separating them from the mesorectum [34]. These layers seem to fuse in the midline, converting DF to a single-layered structure at the center [34]. An interesting conclusion was that DF was easily detached from the mesorectum at the level of SVs and the upper half of the prostate, while it seemed to fuse with the prostatic capsule across the second lower half [34].

Van Ophoven et al. reviewed the literature regarding the anatomy of DF in 1997 and refer to this structure as a macroscopically single-layered fascia with a double-layer configuration at histological examination [35]. Whether these differences in several studies are observed due to various techniques of histological examination and tissue preparation or actually exist due to interindividual variations was recently investigated by Muraoka et al. [36]. They studied 25 male adult cadavers, and after tissue fixation, dissections both at horizontal and sagittal levels were made at 2–5 mm intervals [36]. They detected the existence of collagen along with elastic and smooth muscle fibers in a “leave-like” configuration at the anterior part of DF, while the posterior part was mostly composed of connective tissue [36]. The part of DF covering the surface from SV to mid-prostate was found to have two to eight leaves, which were more clearly defined at the middle-base of SVs and more fragmented at the superior half [36]. The heterogeneity between the various specimens was noted at the level of postero-lateral angle of the prostate and posteriorly to SVs, where half specimens showed a clear and concise lateral connection of DF with EPF while the rest showed an anterior continuation of DF, without a connection to EPF [36]. In the second case, DF partially joined the prostatic capsule laterally and also surrounded the NVB [36]. DF covering the area between mid-prostate and apex partially showed a single-layered configuration, especially at the midline, where DF layers appeared to fuse with the prostatic capsule and the space between the rectum and the prostate was narrow, at less than 3 mm, and without interdisposed loose adipose tissue [36]. A very important feature noted by the authors was the existence of nerve fibers between DF and the prostatic capsule at the midline [36]. The distal border of DF was noted at the superior part of the rectourethralis muscle and inferiorly continued posteriorly to the rhabdosphincter muscle [36].

## 6. DF and Nerve-Sparing Surgery

All these observations are not only interesting from an anatomic point of view but also play an important role during several operative steps of radical prostatectomy or mesorectal excision for rectal cancer. In cases where DF does not fuse with EPF at the posterolateral angle of the prostate, NVB is more dispersed, and both urologists and colorectal surgeons should be careful with the use of wide excision and cautery to avoid traumatizing sensitive nerve fibers. The proposed energy settings to achieve pinpoint coagulation are <30 W cautery level, for a short period of time (<1 s), and ideally avoiding using cautery at close proximity (5–10 mm) with the NVB after pedicle release, as suggested by the Pasadena Consensus Panel for robotic-assisted radical prostatectomy [37].

Costello et al. provided a detailed anatomical description of the NVB after dissecting 12 male cadavers [38]. Pelvic plexus comprises parasympathetic nerve fibers mainly from S4 and at a lesser degree from S2–S3 anterior sacral roots and sympathetic fibers originating from hypogastric nerves bilaterally [38]. The most common location of the pelvic plexus is at the lateral surface of the rectum, above the pararectal fascia and above a layer of 1–2 cm of adipose tissue [38]. Three main branches arise from the pelvic plexus: the anterior branch, coursing across the lateral surface of SVs, the antero-inferior branch, coursing across the lateral prostatic surface and the prostato-vesical junction, and finally, the inferior branch at the postero-lateral prostatic angle [38]. The inferior branch finally forms the NVB with several joining vessel [38]. Similar to Muraoka et al. [36], Costello et al. suggest that DF is increasingly fused with the prostatic capsule at the midline and widens laterally toward the junction with EPF and the pararectal fascia [38]. At this point, NVB fibers course between the several leaves of the interdisposed fascia, and both traction and thermal energy use should be very cautious in nerve-sparing techniques. Tewari et al. proposed the concept of the proximal neural plate, NVB and accessory neural pathways (tri-zonal neural architecture) [39]. The authors reported that the neural plate is located laterally to the bladder neck and SVs at close proximity (5 mm) [39]. This neural structure is sensitive to thermal injury during the dissection of lateral borders of the bladder neck and SVs, as well as while dissecting fascial layers between the prostatic capsule and DF [39]. The NVB is formed as the continuation of the neural plate at the postero-lateral angle of the prostate, as similarly described by Walsh et al. [2,3] and Costello et al. [38]. The authors also suggest the existence of two layers of nerves at the SV base, one superficial between the prostatic capsule and DF and the other coursing within the capsule [39].

Recently, Ghareeb et al. provided detailed anatomic insight regarding the fascia layers of DF and its relationship with NVB [40]. They performed a dissection of 13 male cadavers and described three distinct fascial layers [40]. The most posterior lies in front of the rectal propria fascia and the mesorectal tissue separates these two, while superiorly, this layer extends up to the lowest point of peritoneal reflection and below to the perineal body [40]. The second layer is described to cover SVs along their entire posterior surface and the upper part of the anterior surface up to the semino-prostatic angles [40]. This second layer in all specimens continued to the superior bladder surface and contained a large number of nerve fibers, especially at the 2 and 10 o’clock positions [40]. Therefore, the existence of a large number of NVB fibers between the posterior and intermediate DF layers makes the dissection between them very hazardous for erectile nerves and should be avoided in nerve-sparing procedures [40]. They also describe a third DF layer, which begins at the posterior bladder neck, courses toward the superior bladder surface, and joins with the second layer, while being separated from the bladder fascia with loose connective tissue [40]. This third layer also contains nerve fibers at the 2 and 10 o’clock positions [40]. The authors propose that along the lateral rectal surface, DF fuses with the prehypogastric fascia and rectosacral (Waldeyers’) fascia to form the lateral rectal ligament at the level of S4 [40]. Therefore, they oppose the “H” shaped fascial structure proposed by Kourambas et al. [28] and suggest an inverted “U” shaped architecture [40]. Kiyoshima et al. also confirmed the multilayer theory regarding DF but also described the relationships with the prostatic capsule in 79 non-nerve-sparing RP specimens [26]. They described the fusion between DF and the prostatic capsule in the midline in 97% of cases and the clear distinction of the two structures laterally in 100% of cases [26].

In daily urological surgical practice, the anatomy, number, and topography of several fascial layers around the prostate define the approach to nerve-sparing techniques. Interindividual anatomical variations require a unique surgical dissection in every patient between the proper fascia layers in order to achieve nerve sparing without jeopardizing positive surgical margins [41]. Several grading systems exist to assess nerve sparing, with intrafascial (a surgical plane at the level of prostatic capsule below EPF and DF) and interfascial (between the fascial layers), allowing complete or partial nerve sparing, further depending on anatomical variations and surgical competence [41]. Tewari et al. and Patel et al. have proposed modified grading systems based on veins [42] and arteries [43] along the lateral surface of the prostate as landmarks. Nerve fibers responsible for erectile function can be found between the layers of DF and the prostatic capsule, being more densely packed at the midline of the prostate base and seminal vesicles, while they commonly travel in a more dispersed nature from the postero-lateral border of the prostate toward the apex. During surgery, DF can be recognized as a firm structure, composed of connective tissue after applying tension to SVs, vas deferens, and ventrally pulling the prostate [44] (Figure 3). Although several layers of tissue can be recognized during the histological examination of DF, surgeons face a different scenario intraoperatively. Excessive bleeding, old age associated with tissue changes, adhesions due to previous prostate biopsy, and the lack of magnification commonly lead to the identification of DF as a single-layered tissue macroscopically. As DF fuses with the prostatic capsule at the prostatic base and caudal aspect of SVs (Figure 1), in order to achieve a nerve-sparing procedure without entering the prostatic tissue, an incision should initially be made at the midline of the DF–prostatic capsule interface, and then the dissection should continue laterally on both sides across the layers of DF with a “mesh-like” appearance [36]. In case there is high risk of positive surgical margins, instead of cutting at the junction of DF with the prostatic capsule at the midline, a surgeon can perform a double cut of DF at this location and continue the dissection laterally, as proposed by Martinez-Pineiro [30]. The same plane of dissection continues to the distal border of DF at the prostatic apex [36]. With these two approaches, an intrafascial plane can be achieved (Figure 2). The interfascial plane of dissection refers to cutting of DF at the postero-lateral angle of the prostate (Figure 2), which commonly permits only partial NVB preservation as at this point the NVB is dispersed and the surgeon cuts throughout the nerve fibers [30]. The extrafascial plane (Figure 2) refers to the sacrifice of both the midline fibers and the NVB found at the postero-lateral prostate angle [30].

## 7. DF—Rectal Injury

Rectal injury during RP is a rare event, with the most recent reported rates of incidence being 0.5% in open surgery [45,46], 0.4–2% in laparoscopic RP [47,48], and 0.17–0.3% [49,50] with the robotic technique. Most rectal injuries occur during the dissection of the plane between DF and the rectum from the base of the prostate toward the apex during laparoscopic/robotic technique and the division of rectourethralis muscle and urethra, especially during open surgery [49,51]. Therefore, following the right plane across DF layers after initial incision, either on a retrograde (open surgery) or antegrade (laparoscopic, robotic surgery) approach, while avoiding monopolar coagulation and wide dissection seem to be crucial steps in avoiding this rare but devastating complication [48].

## 8. DF—Colorectal Surgery

Colorectal surgeons should also be familiar with DF as this is the anterior border of total mesorectal excision for rectal cancer operations. Good oncological outcome suggests excision in front of DF layers, but this step jeopardizes NVB integrity and subsequent erectile dysfunction. According to Lindsey et al., anterior dissection can be performed close to the rectum musculature (close rectal plane) to save nerve fibers, between the fascia propria of the rectum and DF (mesorectal plane), or anterior to DF (extramesorectal plane) [16]. Most surgeons seem to follow the mesorectal plane for malignant disease in order to achieve a balance between good oncological and functional outcomes.

## 9. DF—Invasion by Prostate Cancer Cells and Other Clinical Implications

Anatomical planes may offer an entrance or boundary for cancer cell migration. In PCa, the evidence suggests that the transitional zone [52] and prostatic capsule [53] represent a barrier while perineural spaces may become the pathway for cancer spread [53]. The existence of positive surgical margins (PSM) in a specimen of RP is a common occurrence with a reported rate of incidence between 11–48% based on surgical experience and tumor characteristics [54,55]. The presence of PSM leads to a double risk of biochemical recurrence after adjustment for baseline and disease characteristics [56]. The most common location of PSM is the prostatic apex and postero-lateral surface, with 60–75% of PSM observed in these areas, either with open or minimally invasive techniques [57]. DF incision and the creation of a proper dissection plane may help surgeons to identify the safe margin for both maximal nerve sparing and avoiding entering the prostatic capsule, thus avoiding the creation of iatrogenic PSM. Villers et al. demonstrated the significance of the adherence of DF to the prostatic capsule at the midline in contrast with lateral aspects [58]. They report that in 19% of examined specimens, DF was invaded by cancer cells at the midline surface, with this invasion being more prominent in larger tumor volumes (>12 cc) [58]. Interestingly, none of these cases presented with full-thickness DF penetrance, highlighting the protective role of this structure against cancer extension to the rectum [58]. Consequently, great attention should be given while incising DF at the midline as properly conducting this surgical step not only protects from a PSM but also dictates the plane for NVB sparing across the lateral surfaces.

Although DF constitutes a certain physical barrier for the invasion of Pca cells to the rectum, the involvement of this organ has been identified in some patients through lymphatic metastasis or even seeding during transrectal needle biopsy [59]. The risk of direct penetration of PCa to the rectal wall is more commonly observed in patients with a large tumor burden and also in those with neoplasm located in the prostate central zone or the gland’s base [58]. The most reasonable explanation for this observation is that DF and the prostate are not separated by elastic or adipose tissue at the midline, making invasion from aggressive tumors easier [58]. Invasion to the rectal wall is commonly associated with the extended extra-prostatic extension of neoplasm as a sign of advanced disease stage. Although surgery in these patients has not been a promising way of management in the past, Wang et al. suggest that total pelvic exenteration can be applied in cases of patients previously not treated with hormonal therapy, followed by postoperative androgen deprivation therapy, with some patients achieving long-term survival [60].

DF manipulation in the form of expansion using a polyethylene glycol hydrogel, which was injected into DF under ultrasound guidance, was tested by Abreu et al. [61]. The authors reported an uneventful hydrogel instillation into DF and an effective expansion of the fascia, which was maintained for one hour and aided in maintaining a proper temperature during cryoablation focal therapy for PCa [61].

## 10. DF and Urinary Continence

A terrifying complication of RP is urinary incontinence, which can persist in up to 21% of patients at 12 months post-operation, regardless of the surgical approach [62]. Several operative steps and maneuvers have been described to achieve continence, including preservation of urethral sphincter, adequate urethral length, reconstruction of bladder neck, nerve-sparing procedures, and also avoid the removal of DF [41]. This fascia becomes denser at the midline, commonly fusing with the prostate to form dorsal raphe [26]. This tendinous structure continues from the base to the apex of the prostate and is considered to support the urethra and prostate as a fulcrum [63]. The rest of DF across the posterior prostatic surface is considered to act as a hammock to support vesicourethral anastomosis [64]. Finally, the scattered neural fibers located in NVB are dispersed along the multiple DF layers and are responsible for the innervation of both corpora cavernosa and urethral sphincter [41]. Recently, Lu et al. performed a comparison between patients who had DF sparing during RP versus those who did not undergo DF-sparing surgery due to intraoperative and oncological reasons [64]. They found that immediate continence was statistically and clinically higher than those without DF sparing (83.3% versus 13.4% at 1 week postoperatively, *p* < 0.01) [64]. The sparing of DF also led to improved continence rates at the end of the 12th postoperative month and improved potency rates (34.7% vs. 17.1%, *p* = 0.01) while positive surgical margins did not differ significantly between the two groups [64]. From a clinical point of view, recognizing and preserving DF without compromising oncological safety seems to lead to better potency and continence rates. Again, this can be achieved by cutting in front of the anterior layer of DF at the midline and continuing in the same way across lateral prostatic surfaces. Blunt dissection at this point is preferable because the application of thermal energy can damage sensitive nerve fibers.

## 11. Conclusions

DF is a multilayered fascial structure covering the posterior surface of the prostate from the base to the apex and lateral surfaces where it joints with EPF and pararectal fascia. The identification of the proper entry site in this fascia during RP facilitates the avoidance of PSM and rectal injury, while it is an essential step to maximize nerve sparing at the level of SVs and postero-lateral NVB. From a clinical point of view, DF is a critical barrier to the spread of PCa cells toward the rectal wall and lumen, while its manipulation can protect the rectum during focal ablative therapies. For these reasons, both urologists and colorectal surgeons should familiarize themselves with DF anatomy and approaches during surgery.

## Figures and Tables

**Figure 1 cancers-14-00688-f001:**
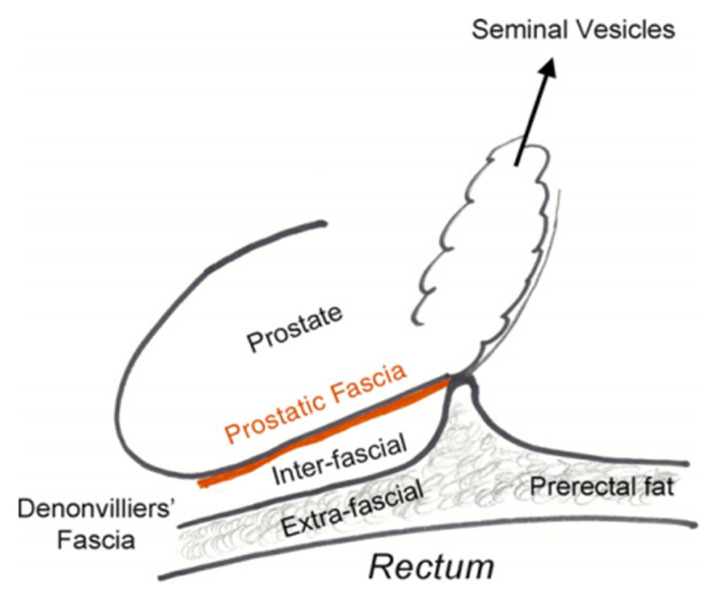
Prostatic fasciae anatomy. Reprinted with permission from [30]. Copyright 2021 Elsevier.

**Figure 2 cancers-14-00688-f002:**
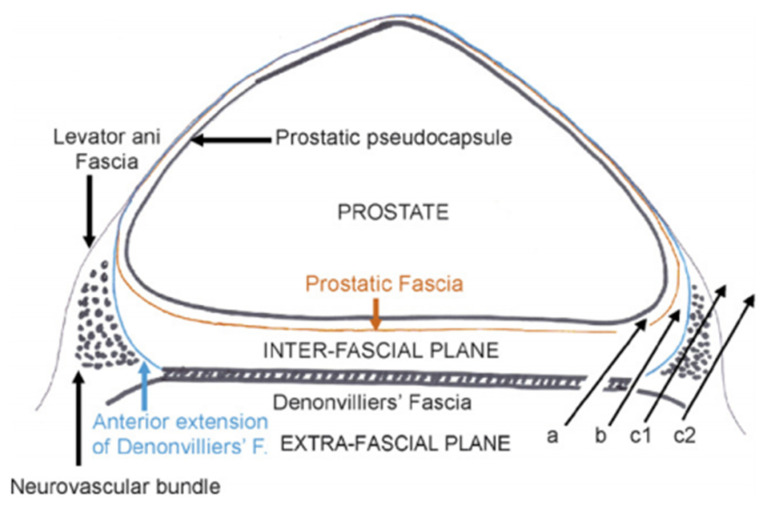
Surgical planes for nerve-sparing radical prostatectomy. a = intrafascial plane; b = interfascial plane; c1, c2 = extrafascial plane. Reprinted with permission from [30]. Copyright 2021 Elsevier.

**Figure 3 cancers-14-00688-f003:**
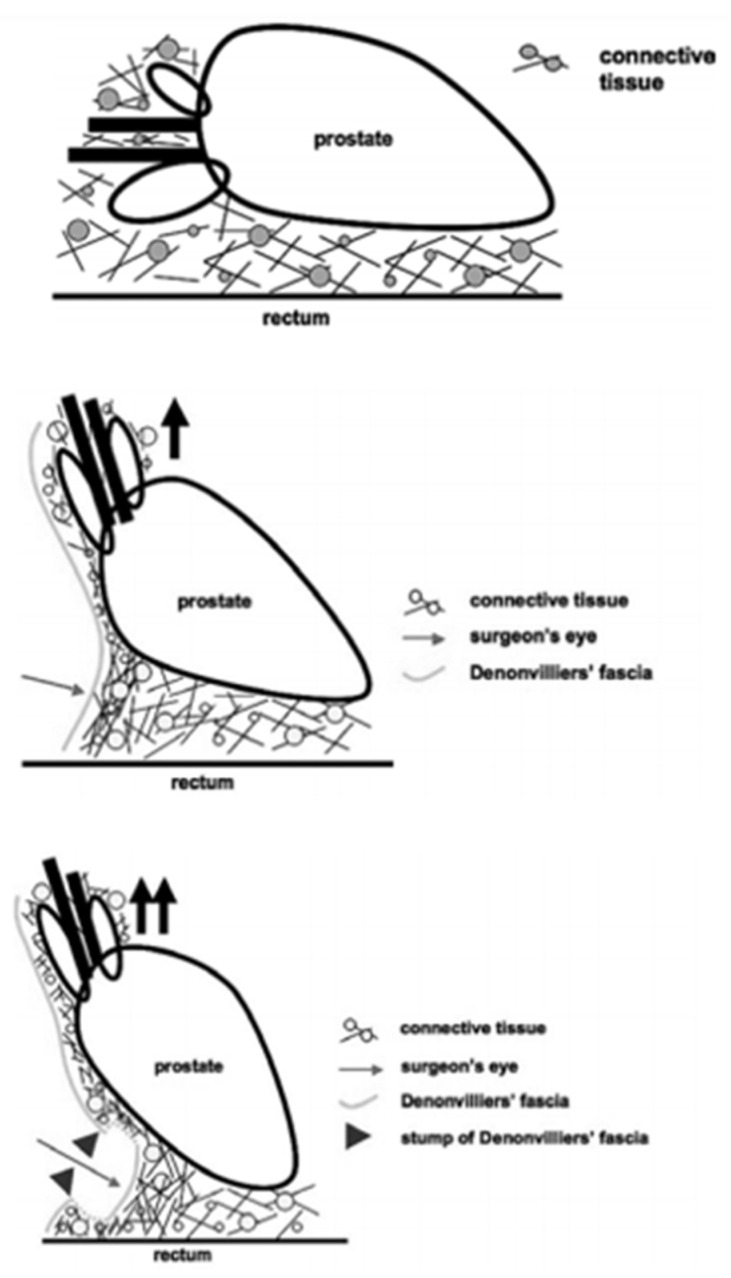
Intraoperative identification of Denonvilliers’ fascia by pulling the prostate ventrally. Reprinted with permission from [44]. Copyright 2021 Spandidos Publications.

## Data Availability

Data sharing not applicable.

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
