# Peer review of "Denonvilliers’ Fascia: The Prostate Border to the Outside World"

_cancers, 2022, doi:10.3390/cancers14030688_

Round 1
Reviewer 1 Report
Tzelves et al. wrote a comprehensive review on the fasciae surrounding the prostate - with a focus on Denovilliers´ fascia - describing its embryonic development, anatomy and clinical/surgical implications. The paper is well-written and very interesting especially for urologic surgeons performing radical prostatectomies. However, the manuscript could be improved by the following measures:
- The english language could be markedly improved, I would suggest the authors to send the manuscript to an English editing service, like the one of MDPI.
- The structure of the manuscript is a bit confusing, for example, there is a subchapter 5.1 but no following ones (like 5.2 or 5.3). Please reconsider the structure of the manuscript to make it easier to read and avoid redundancies.
- Some intraoperative pictures of endopelvic or Denoviellers´ fascia would be very interesting and helpful for the readers.
Author Response
Dear Reviewer 1,
We want to express our gratitude for putting effort to review our manuscript and make these comments, which certainly helped to improve its quality and readability for Journal’s readers. We proceeded with appropriated changes/ additions to our manuscript according to your comments and all can be found both within the text and also as answers to specific questions.
With kind regards,
Lazaros Tzelves
- ‘’The english language could be markedly improved, I would suggest the authors to send the manuscript to an English editing service, like the one of MDPI.’’
Answer:
We want to thank Reviewer for effort in reviewing our manuscript and for the proposal to improve English language. We have proceeded accordingly and an English native speaker has made appropriate changes within the manuscript.
- ‘’The structure of the manuscript is a bit confusing, for example, there is a subchapter 5.1 but no following ones (like 5.2 or 5.3). Please reconsider the structure of the manuscript to make it easier to read and avoid redundancies.’’
Answer:
We wish to thank the Reviewer for noticing this inconsistency to the numbering and structure of our text. We have removed subsection 5.1 according to the comment.
- Some intraoperative pictures of endopelvic or Denoviellers´ fascia would be very interesting and helpful for the readers.
Answer:
We wish to thank the Reviewer for this comment. We totally agree that a clear intraoperative picture of Denonvilliers’ fascia or Endopelvic fascia would be very helpful. However we were not able to grant a relevant permission from online texts and since our Hospital has been turned to a Unit only for COVID-positive patients, currently we do not perform operations in order to acquire such intraoperative images during radical prostatectomy. Therefore, we apologize for not being able to provide such images although we totally understand how useful they would be.
Reviewer 2 Report
The present article is a simple and linea anatomic description of some peri prostatic elements such as Denonvilleir fascia.
It is not able to give information particularly helpful for a clinical point of view.
In particular if the topic was Denonvilleir fascia, this structure could be better investigated on the following points:
1: periprostatic innervation and their relationship with the fascia
2. better description of the different layer of the fascia
3. different approaches in radical prostatectomy related to the fascia
4. better description on the risk of oncologic invasion of the fascia prom prostate and rectal cancers
5. possible relationships between the fascia and urinary continence
In general authors should try to transform this simple anatomic description in some of larger clinical interest for readers
Author Response
Dear Reviewer 2,
We want to express our gratitude for putting effort to review our manuscript and make these comments, which certainly helped to improve its quality and readability for Journal’s readers. We proceeded with appropriated changes/ additions to our manuscript according to your comments and all can be found both within the text and also as answers to specific questions.
With kind regards,
Lazaros Tzelves
- ‘’In particular if the topic was Denonvilleir fascia, this structure could be better investigated on the following points:
1: periprostatic innervation and their relationship with the fascia
- better description of the different layer of the fascia
- different approaches in radical prostatectomy related to the fascia
- better description on the risk of oncologic invasion of the fascia prom prostateand rectal cancers
- possible relationships between the fascia and urinary continence
In general authors should try to transform this simple anatomic description in some of larger clinical interest for readers’’
Answer:
We wish to express our gratitude for these observations and comments from Reviewer 2, which certainly show the understanding and effort made to read our manuscript. We have added several sentences and sections to cover all these very important clinical points. These were added within the manuscript either as additions to pre-existing sections or as new sections. Specifically to answer point 1,2 and 3 we have modified the clinical points given in section 6 (DF and nerve-sparing surgery),which now reads as follows: ‘’ In daily urological surgical practice, the anatomy, number and topography of several fascial layers around the prostate, define the approach to nerve-sparing techniques. Inter-individual anatomical variations require a unique surgical dissection in every patient between the proper fascia layers, in order to achieve nerve-sparing without jeopardizing positive surgical margins[41]. Several grading systems exist to assess nerve-sparing, with intrafascial (a surgical plane at the level of prostatic capsule below EPF and DF) and interfascial (between the fascial layers), allowing complete or partial nerve-sparing, depending also on anatomical variations and surgical competence[41]. Tewari et al and Patel et al have proposed modified grading systems, based on veins[42] and arteries[43] along the lateral surface of prostate, as landmarks. Nerve fibers responsible for erectile function can be found between the layers of DF and prostatic capsule, being more densely packed at the midline of prostate base and seminal vesicles, while they commonly travel more dispersed from the postero-lateral border of prostate towards the apex. During surgery, DF can be recognized as a firm structure, composed of connective tissue after applying tension to SVs, vas deferens and pulling prostate ventrally[44] (Figure 3). Although several layers of tissue can be recognized during histological examination of DF, surgeons face a different scenario intraoperatively. Excessive bleeding, old age associated with tissue changes, adhesions due to previous prostate biopsy and the lack of magnification commonly leads to identification of DF as a single-layered tissue macroscopically. Since DF fuses with prostatic capsule at the prostatic base and caudal aspect of SVs (Figure 1), in order to achieve a nerve-sparing procedure without entering prostatic tissue, an incision should initially be made at the midline of DF-prostatic capsule interface and then dissection should continue laterally on both sides across the layers of DF with the ‘’mesh-like’’ appearance[36]. In case there is high risk for positive surgical margins, instead of cutting at the junction of DF with prostatic capsule at the midline, surgeon can perform a double cut of DF at this location and continue dissection laterally, as proposed by Martinez-Pineiro[30]. The same plane of dissection continues to the distal border of DF at the prostatic apex[36]. With these two approaches, an intrafascial plane can be achieved (Figure 2). Interfascial plane of dissection refers to cutting of DF at the postero-lateral angle of the prostate (Figure 2), which commonly permits only partial NVB preservation, since at this point the NVB is dispersed and surgeon cuts throughout the nerve fibers[30]. Extrafascial plane (Figure 2) refers to sacrifice of both the midline fibers and also the NVB found at the postero-lateral prostate angle[30].’’
Regarding point 4, we have added the following text to section 9 (DF – Invasion by prostate cancer cells and other clinical implications):’’ The risk of direct penetration of PCa to rectal wall is more commonly observed in patients with large tumor burden and also at those with neoplasm located in prostate central zone or gland’s base[58]. The most reasonable explanation for this observation is that DF and prostate are not separated by elastic or adipose tissue at the midline, making invasion from aggressive tumors easier[58]. Invasion to rectal wall is commonly associated with extended extra-prostatic extension of neoplasm, as a sign of advanced disease stage. “, which comes from the observations of Villers and McLean about DF invasion from prostate cancer cells and explain which cases are in greater danger of experiencing this type of extraprostatic extension.
Regarding point 5, we have added the following section: ‘’ 10. DF and urinary continence: A terrifying complication of RP is urinary incontinence, which can persist in up to 21% of patients at 12 months postoperatively, regardless of surgical approach[62]. Several operative steps and maneuvers have been described to achieve continence, including preservation of urethral sphincter, adequate urethral length, re-construction of bladder neck, nerve-sparing procedures and also avoid removal of DF[41]. This fascia, becomes denser at the midline, commonly fusing with prostate to form dorsal raphe[26]. This tendinous structure continues from base to apex of the prostate and is considered to support urethra and prostate as a fulcrum[63]. The rest of DF across posterior prostatic surface, is considered to act as a hammock to support vesicourethral anastomosis[64]. Finally, the scattered neural fibers located in NVB, are dispersed along the multiple DF layers and they are responsible for innervation of both corpora cavernosa and urethral sphincter[41]. Recently, Lu et al performed a comparison between patients who had DF-sparing during RP versus those who did not undergo a DF-sparing surgery due to intraoperative and oncological reasons[64]. They found that immediate continence was statistically and clinically higher than those without DF-sparing (83.3% versus 13.4% at 1 week postoperatively, p<0.01)[64]. Sparing of DF, led also to improved continence rates at the end of 12th postoperative month, improved potency rates (34.7% vs 17.1%, p=0.01), while positive surgical margins did not differ significantly between the two groups[64]. From a clinical point of view, recognizing and preserving DF, without compromising oncological safety, seems to lead to better potency and continence rates. Again this can be achieved by cutting in front of the anterior layer of DF at the midline and continue the same way across lateral prostatic surfaces. Blunt dissection at this point is preferable, since application of thermal energy can damage sensitive nerve fibers. ‘’ The following references were also added: 62. Haglind, E.; Carlsson, S.; Stranne, J.; Wallerstedt, A.; Wilderäng, U.; Thorsteinsdottir, T.; Lagerkvist, M.; Damber, J.E.; Bjartell, A.; Hugosson, J.; et al. Urinary Incontinence and Erectile Dysfunction After Robotic Versus Open Radical Prostatectomy: A Prospective, Controlled, Nonrandomised Trial. Eur Urol 2015, 68, 216-225, doi:10.1016/j.eururo.2015.02.029., 63. Dalpiaz, O.; Anderhuber, F. The fascial suspension of the prostate: A cadaveric study. Neurourology and urodynamics 2017, 36, 1131-1135, doi:10.1002/nau.23073., 64. Lu, X.; He, C.; Zhang, S.; Yang, F.; Guo, Z.; Huang, J.; He, M.; Wu, J.; Sheng, X.; Lin, W.; et al. Denonvilliers’ fascia acts as the fulcrum and hammock for continence after radical prostatectomy. BMC Urology 2021, 21, 176, doi:10.1186/s12894-021-00943-z.
Reviewer 3 Report
Authors described some fascias surrounding prostate, especially detailed in Denonvilliers' fascia (DF). DF is one of the most important anatomical structure to conduct radical prostatectomy (RP). Urologist should have precise and profound understandings of DF to achieve sufficient surgical and oncological outcome in RP. Thus, this work could be beneficial to summarize DF's embryological origin and detailed structures.
Manuscript was made in well written contents and English connection.
I realized only minor revision points.
#1: In line 29, Space is wide between "yet." and "A number of".
#2: In line48, the aim of this study is to review the anatomy of this fascia and provide insights for specific steps during RP.
#3 After line 265, "Figure 8." and "Figure 9." are inserted without any explanation. These should be removed.
Author Response
Dear Reviewer 3,
We want to express our gratitude for putting effort to review our manuscript and make these comments, which certainly helped to improve its quality and readability for Journal’s readers. We proceeded with appropriated changes/ additions to our manuscript according to your comments and all can be found both within the text and also as answers to specific questions.
With kind regards,
Lazaros Tzelves
- ‘’In line 29, Space is wide between "yet." and "A number of".’’
Answer:
We wish to thank the Reviewer for noticing this extra space. We addressed it accordingly.
- ‘’In line48, the aim of this study is to review the anatomy of this fascia and provide insights for specific steps during RP.’’
Answer:
We wish to thank the Reviewer for noticing this lack of ‘is’. We modified the sentence according to a language revisions and now reads as follows: ‘’ The aim of this review is to present the anatomy of this fascia and provide insights for specific steps during RP.’’
- ‘’After line 265, "Figure 8." and "Figure 9." are inserted without any explanation. These should be removed.’’
Answer:
We wish to thank the Reviewer for noticing this inconsistency of Figure 3. We have modified the figure accordingly.
Round 2
Reviewer 2 Report
After revision, the manuscript has been implemented and can be considered useful for readers and suitable for publication